# OpenReview forum: "XtraGPT: LLMs for Human-AI Collaboration on Controllable Scientific Paper Refinement"
_ICLR.cc/2025/Workshop/BuildingTrust — Submitted to BuildingTrust_

### Official Review · Reviewer_7dvi · 2025-02-28

**Rating:** 7
**Confidence:** 4

**Review:**

The paper introduces and explores a pretty interesting topic of leveraging large models for academic writing. It would be of certain contribution to have open-source models comparable to closed-source models in academic writing for further research study.

Strength:
- The idea of finer level control for generation is of certain importance. The authors define a clear flow of how we show guide LLM for generation
- The experiments and analysis propose some interesting findings.

Weakness:
- For the first question in the paper, "Why Can’t Existing LLMs Excel in Paper Generation?", Apart from the reasons raised in the paper, one potential drawback of existing tools is that they cannot read tables and figures, nor generate tables or figures. However, "Figures and tables are key sources of information in many scholarly documents."[1]. As a result, a potential weakness of the paper is that this critical issue is not discussed nor resolved though it attempts to figure out the reason why LLMs cannot excel in paper generation.
- The writing is more like a blog rather than an academic though the paper is trying to study the scheme of academic papers. For example, the quote at the beginning of the introduction is not of a certain importance and is pretty weird.

[1]Clark, Christopher, and Santosh Divvala. "Pdffigures 2.0: Mining figures from research papers." In Proceedings of the 16th ACM/IEEE-CS on Joint Conference on Digital Libraries, pp. 143-152. 2016.

---

### Official Review · Reviewer_deWm · 2025-03-01
**Mediocre results and clarifications needed to justify the paper's contributions**

**Rating:** 4
**Confidence:** 4

**Review:**

Strengths:
- This paper introduces a fine-tuned model that can generate scientific writing and its training dataset.
- The released model is shown to be able to outperform existing models on the task.

Weaknesses:
- Are there words missing in line 18? The grammar seems weird in this sentence “The XtraGPT training and evaluation processes, and models will be open-sourced”. There are also quite some more grammar errors in the paper, which makes it hard to understand the paper sometimes.
- It seems like the training data was generated using GPT-4o-mini as described at line 188, but it is unclear whether the authors had done some quality checks over the generated data as the model can still hallucinate given the nonzero hallucination rate. It’ll be helpful if the authors can clarify this or include an agreement rate of the model’s generated data against human’s preference to establish more trust in their curated dataset. The authors seem to mention that some results on this are included in table 8 and table 9, but the captions of those tables were confusing, making it difficult to interpret the results without any more explanation in text.
- Visibility of Figure 2 should be improved.
- The paper only focused on one data distribution – 2024 ICLR papers. Since XtraGPT is proposed as a general scientific writing model, I wonder how the results shown in the paper can generalize to different distributions, especially when the model is trained using a technique like SFT. The authors should include some testing on unseen distributions to provide more practical insights.
- I’m not sure how significant the result is. The biggest contribution claimed in the paper is the model’s ability to supposedly help refine a paper. To demonstrate, the authors claimed that randomly “refining” one paragraph from each of the six sections in a paper using their model can improve the overall rating by 0.02, but I’m not sure whether an improvement of 0.02 in the overall rating can be called significant or not.
- I’d like to see a more in-depth analysis on which aspects the model outperforms human’s writing. For example, is it that the model generates text with better grammar? a better flow? or a more integrated discussion that better synthesizes important information from the context? This can help provide more useful actionable insights for users to know which sections/parts are the most useful to leverage AI to help. However, the paper only evaluated the model on papers undergone random replacement of paragraphs, which don’t really provide much actionable insights.

---

### Official Review · Reviewer_q4jg · 2025-03-02
**Meaningful research topic with unclear paper writing**

**Rating:** 3
**Confidence:** 4

**Review:**

**Summary**:
This paper presents the problem of how to use the LLM to assist humans to write academic papers. To solve this problem, authors collected a dataset called XtraQA of high-quality question-answer pairs. Additionally, they proposed a model, XtraGPT, of 7B model size to help paper writing. Their results show that the proposed model surpass the models of the similar size and be comparable to GPT-4o-mini.

**Strengths:**
  1. The research topic (i.e., how to leverage LLMs to help academic paper writing) is interesting and meaningful.

**Weaknesses:**
  1. The overall writing of the whole paper cannot achieve the level of an academic paper, which happens to be the topic of this paper (i.e., how to use LLMs to help paper writing). I don't believe this paper is ready to publish or is even ready for reviewers to review.
  2. Specially, I can point out several evidences:
        (1) The first paragraph of the introduction section is unnecessary.
        (2) In the abstract, the paper mentioned "The XtraGPT training and evaluation ...." (Line 17-18) even before mentioned the concept of XtraGPT (Line 19).
        (3) In Line 072, it mentioned the controllable generation, but the meaning of it is not clearly and explicitly explained in the paper.
        (4) In 085, the enumerate number "3)" was forgotten.
        (5) In Line 309 to 323, there is no intuition and explanation why the LC win rate is defined as this way. It is invented by the authors, or it is a predefined concept?
        (6) Figure 4 have overlapping x ticks for all four subplots, which makes it hard to see them clearly.
  3. After reading the whole paper, it is not very clear to me why I need to use XtraGPT to assist myself to refine my paper, and what its advantages compared to using ChatGPT (e.g., GPT-4o) for this purpose?
  4. The methodology used is not very sound. The paper is using LLM-as-a-Judge evaluation, but it doesn't discuss what the limitations (e.g., length bias, position bias, etc.) are for employing it and how to mitigate these limitations.

---

### Decision · Program_Chairs · 2025-03-04

**Decision:**

Reject

**Comment:**

Given the low relevance to the workshop and the weaknesses outlined by R1 and R2, I recommend rejection.